# Isolation and Identification of *Alternaria alternata* from Potato Plants Affected by Leaf Spot Disease in Korea: Selection of Effective Fungicides

**DOI:** 10.3390/jof10010053

**Published:** 2024-01-07

**Authors:** Jiyoon Park, Seoyeon Kim, Miju Jo, Sunmin An, Youngjun Kim, Jonghan Yoon, Min-Hye Jeong, Eun Young Kim, Jaehyuk Choi, Yangseon Kim, Sook-Young Park

**Affiliations:** 1Department of Plant Medicine, Sunchon National University, Sunchon 57922, Republic of Korea; jiyoonpak@gmail.com (J.P.); cbc6976@gmail.com (S.K.); miju7188@gmail.com (M.J.); sminan201@gmail.com (S.A.); nate9702@gmail.com (Y.K.); yoonjonghan99@gmail.com (J.Y.); minhye1962@gmail.com (M.-H.J.); 2Interdisciplinary Program in IT-Bio Convergence System (BK21 Plus), Sunchon National University, Suncheon 57922, Republic of Korea; 3Crop Cultivation and Environment Research Division, National Institute of Crop Science, Rural Development Administration, Suwon 16429, Republic of Korea; key4082@korea.kr; 4Department of Life Sciences, College of Life Sciences and Bioengineering, Incheon National University, Incheon 22012, Republic of Korea; jaehyukc@inu.ac.kr; 5Department of Research and Development, Center for Industrialization of Agricultural and Livestock Microorganisms, Jeongeup-si 56212, Republic of Korea; yangseon@cialm.or.kr

**Keywords:** *Alternaria alternata*, brown spot disease, pathogenicity test, phylogenetic analysis, *Solanum tuberosum* L.

## Abstract

Brown leaf spot disease caused by *Alternaria* spp. is among the most common diseases of potato crops. Typical brown spot symptoms were observed in commercial potato-cultivation areas of northern Korea from June to August 2020–2021. In total, 68 isolates were collected, and based on sequence analysis of the internal transcribed spacer (ITS) region, the collected isolates were identified as *Alternaria* spp. (80.9%). Phylogenetic analysis revealed that a majority of these isolates clustered within a clade that included *A. alternata*. Additionally, the ITS region and *rpb2* yielded the most informative sequences for the identification of *A. alternata*. Pathogenicity tests confirmed that the collected pathogens elicited symptoms identical to those observed in the field. In pathogenicity tests performed on seven commercial cultivars, the pathogens exhibited strong virulence in both wound and non-wound inoculations. Among the cultivars tested, Arirang-1ho, Arirang-2ho, and Golden Ball were resistant to the pathogens. Furthermore, among the fungicides tested in vitro, mancozeb and difenoconazole were found to be effective for inhibiting mycelial growth. In summary, our findings suggest that *A. alternata* plays a critical role in leaf disease in potato-growing regions and emphasise the necessity of continuous monitoring and management to protect against this disease in Korea.

## 1. Introduction

Potatoes (*Solanum tuberosum* L.) are a staple non-cereal food crop and are the fourth-most-productive crop after maize, wheat, and rice worldwide [1]. Various pathogenic infections threaten potato crops and can lead to poor quality and reduced yield. These infections include late blight, caused by *Phytophthora infestans*; early blight, caused by *Alternaria solani*; and brown leaf spot disease, caused by *Alternaria alternata* [2]. In a previous study conducted in Korea, more than 60% of potato leaves exhibited brown leaf spots caused by *A. alternata* infection under conditions of high humidity and warm temperatures that ranged from 12 °C at night to 30 °C during the daytime [3].

Unlike early blight, potato brown leaf spot disease caused by *A. alternata* manifests as small, irregular, circular spots on the lower leaves, with sizes ranging from pinpoint to 12 mm, that turn into dark brown spots without concentric rings [4]. This disease occurs throughout the potato-growing season when environmental conditions are favourable for the pathogen. As the disease progresses, the entire leaf becomes chlorotic and turns brown and the leaf edges curl up, mirroring the symptoms of early blight. Eventually, the infected leaves wither and die and are left hanging from the potato plant. Dorby et al. (1984) reported that yields could be reduced by as much as 18% under conditions of high relative humidity and temperature, high pathogen density, and host susceptibility [5].

This disease has been relatively underresearched in comparison to early and late blight, two other diseases of potato plants. However, a brown spot disease caused by *A. alternata* has been recently reported in several countries [1,3,6,7,8]. Previous studies have reported that *A. arborescens*, *A. tenuissima*, *A. tomatophila*, *A. grandis*, *A. solani*, and *A. alternata* are the causal pathogens of brown leaf blight symptoms in potato plants [1,9,10,11].

The genus *Alternaria* is difficult to identify based solely on morphology, and employing the commonly used internal transcribed spacer (ITS) region for sequence-based identification remains challenging. Phylogenetic analysis using concatenated sequences offers a potential solution for resolving fungal species classification within the genus [11,12]. Woudenberg et al. (2015) generated sequences from seven loci: the ITS region, glyceraldehyde-3-phosphate dehydrogenase (*gapdh*), translation elongation factor 1-alpha (*tef1*), RNA polymerase second-largest subunit (*rpb2*), *Alternaria* major allergen gene (*Alt a 1*), endopolygalacturonase (*endoPG*), and OPA10-1 [11]. Methods beyond traditional morphological and phylogenetic analyses that focus on these seven loci would aid in the definitive identification of *Alternaria* species.

Identifying the pathogen causing a disease is the first step towards its control. In a previous study, we reported that *A. alternata* is the causal pathogen of brown leaf spot symptoms observed during the harvest season in the potato-cultivation areas of Yeoncheon, Gyeonggi Province, Korea [3]. In this study, we employed a comprehensive set of sequences corresponding to seven loci, namely ITS, *gapdh*, *tef1*, *rpb2*, *Alt a1*, *endoPG*, and OPA10-1, and generated a phylogenetic tree. This approach allowed us to identify the causal pathogen as *A. alternata* [3]. Developing resistant potato cultivars is the most efficient strategy for effective control. However, as for potato early blight, no cultivars resistant to potato brown leaf spot have been reported to date [13]. Therefore, implementing efficient fungicide application will help mitigate the reduction in potato yield caused by this pathogen [14].

The objectives of this study were as follows: (i) to use molecular analyses to identify the predominant causal pathogen responsible for leaf blight symptoms, including leaf spots, in leaves collected from Yeoncheon, Baengnyeongdo, and Goseong between 2020 and 2021 and, if dominant pathogens were identified; (ii) to analyse the differences among the dominant species via phylogenetic analysis; (iii) to examine their pathogenicity in different potato cultivars; and (iv) to identify effective fungicides. In this study, we identified *A. alternata* as the primary pathogen responsible for leaf spot and blight symptoms in samples collected from the three regions from June to August 2020–2021. Furthermore, we performed a phylogenetic analysis using seven loci and found that the ITS and *RPB2* loci are the most informative sequences for distinguishing this species from others in the genus and for distinguishing among strains of *A. alternata*.

## 2. Materials and Methods

### 2.1. Isolation of Fungal Isolates

Potato leaves that had developed symptoms were collected from the ‘Superior’ cultivar in three different regions from June to August 2020–2021. Infected tissues (5 × 5 mm) from the diseased leaf samples were immersed in 70% ethanol for 1 min, rinsed three times in sterilised water, dried, placed on water agar amended with 100 g/mL of streptomycin, and then incubated in the dark at 25 °C for 3–7 days. After hyphae emerged from the tissues, the fungal isolates were transferred onto potato dextrose agar (PDA; Difco Laboratories, Detroit, MI, USA) or V8-Juice agar medium (8% Campbell’s V8-Juice, 1.5% agar, pH adjusted to 7 using 0.1 N NaOH). All fungal strains were stored at 4 °C in sterile distilled water or were placed in long-term storage until the experiment at −80 °C in 15% glycerol in the agar blocks on which the fungi were grown.

### 2.2. Fungal Cultures and DNA Extraction

All the collected isolates were subjected to DNA extraction. Fungal isolates were grown in 5 mL of potato dextrose broth (Difco Laboratories, Detroit, MI, USA) at 25 °C for 7 days. Genomic DNA was isolated as described previously [15] or purified using the NucleoSpin Plant kit (Macherey-Nagel, Dűren, Germany) according to the manufacturer’s instructions. The DNA concentration was estimated using a NanoDrop ND-1000 spectrophotometer (NanoDrop Technologies, Inc. Wilmington, NC, USA). The DNA concentration was adjusted to 12.5 ng/μL for each isolate and subjected to PCR amplification.

### 2.3. PCR and Sequencing

PCR was performed using an ABI 2720 Thermal Cycler (Applied Biosystems, Foster City, CA, USA). PCR amplification was performed with 25 ng of genomic DNA and 2 pmol/L of each primer (Table 1) using the i-StarMAX II PCR master mix (iNtRON Biotechnology Inc., Seongnam, Republic of Korea). The amplification conditions were as follows: (a) initial denaturation at 96 °C for 1 min; (b) 2 cycles of denaturation at 94 °C for 1 min, annealing at 52 °C for 1 min, and elongation at 72 °C for 1 min; (c) 28 cycles of denaturation at 94 °C for 30 s, annealing at 55 °C for 30 s, and elongation at 72 °C for 1 min; and (d) elongation at 72 °C for 3 min.

PCR products were resolved via 0.8% agarose gel electrophoresis and bidirectionally sequenced by the Bioneer sequencing service (Bioneer Inc., Daejeon, Republic of Korea) on both strands with the same primers used for PCR amplification. Sequence assembly was performed using the SeqMan program DNAStar (Madison, WI, USA) and CodonCode Aligner V3.5.4. software (CodonCode Co., Centerville, MA, USA). The aligned sequences were subjected to a BLASTn search in the GenBank database “http://www.ncbi.nlm.nih.gov/BLAST (accessed on 1 December 2023)”. All of the generated sequences were deposited in GenBank (Appendix A).

### 2.4. Phylogenetic Analysis

Phylogenetic analysis was performed on *A. alternata* isolates, and the dominant group was identified using ITS region sequencing. Multiple sequence alignments of the concatenated sequences were generated using ClustalX [24] and manually adjusted. Sequence divergence was estimated using the MEGA computer package version 11 [25] and the Tamura-Nei model of evolution [26]. Phylogenetic analyses of the sequence data consisted of a maximum-likelihood analysis of both the individual data partitions and the combined dataset.

### 2.5. Pathogenicity Test

To fulfil Koch’s postulates and confirm that these fungi could infect potato leaves, 1-month-old potato plants (*S. tuberosum* cultivar (cv.) Superior) grown in a 25 °C growth chamber were sprayed with a conidial suspension (1 × 10^6^ conidia/mL) containing 250 ppm Tween 20 prepared from 7-to-14-day-old cultures of the selected *Alternaria* spp. isolates. Sterile distilled water was used as the control. The inoculated plants were placed in a plastic box (50 × 40 × 45 cm) to maintain high humidity and incubated in the dark at 25 °C for 1 day. The box was transferred to a growth chamber, and the plants were grown under a 16-h photoperiod with fluorescent lighting and maintained at a temperature of 25 °C and humidity >70%. Disease severity was measured 7 days after inoculation. The assay was performed in triplicate.

### 2.6. Virulence Test on Commercial Cultivars

Seven commercial potato cultivars, Arirang-1ho, Arirang-2ho, Golden Ball, Daekwang, Daeji, Superior, and Chubaek, were obtained from the Highland Agriculture Research Institute, National Institute of Crop Science, Rural Development Administration, Korea. The cultivars were grown in a greenhouse (23–30 °C). For large-scale screening, a detached leaf assay was performed using leaves from 45-day-old plants from all seven potato cultivars.

Healthy leaves were placed in a plastic box and maintained in a watered state using cotton. Then, a single leaf was inflicted with wounds 10 times at each of the three inoculation sites using a hand acupuncture needle (0.18 × 8 mm, Qingdao Dongbang Medical Co., Ltd., Shandong, China). The inoculum was prepared with 6-mm agar plugs from the 7-day-old *A. alternata* culture in a V8 juice agar medium. For inoculation, the mycelial agar plugs were placed upside down onto the detached leaves.

Inoculation was conducted in the dark at 25 °C. At 1 day post-inoculation (dpi), the inoculum was removed from the infection sites and the containers were incubated in a growth chamber (16 h light with >70% humidity and at 25 °C). The symptoms were observed at 7 dpi. An uninoculated V8 juice–agar plug was used as a control. All the experiments were performed twice. Diseased leaf area (DLA) was calculated by measuring the leaf area with observable symptoms relative to the total observed area using ImageJ software version 1.48 [27]. The formula used was as follows:Diseased leaf area (%) = (leaf area with visible symptoms/total observed area) × 100.(1)

### 2.7. In Vitro Screening of Fungicide Sensitivity

The ability of the fungicides to inhibit the radial growth of the *A. alternata* isolates was assayed. To select a suitable fungicide for the control of *A. alternata* isolates, different fungicides with varied mechanisms of action were tested: mancozeb and chlorothalonil; difenoconazole; boscalid and pydiflumetofen; kresoxim-methyl; and thiophanate-methyl (Table 2).

After the *A. alternata* isolates were selected, the ability of the fungicides to inhibit the radial growth of these isolates was evaluated using the agar dilution method [28]. The selected fungicide was added to the PDA medium before the medium solidified, and the mycelial plug from the edge of the hyphae was cultured for 7 days in the PDA medium using a 6 mm cork borer. After 7 days, the mycelial plugs were inoculated into the fungicide medium.

After 7 days of incubation at 25 °C, the amounts of radial growth in both the control (C, PDA) and treated (T, PDA amended with fungicide) groups were measured. The percentages of radial growth inhibition (I) and corrected inhibition (IC) were calculated as previously described [29]. In brief, two formulae were used:I (%) = [(C − T)/C] × 100 (2)
IC (%) = [(C − T)/(C − C_0_)] × 100(3)
where C is the diameter of the fungal colony from the selected isolate on the PDA plate, T is the diameter of the colony on the treated plate, and C_0_ is the diameter of the primary fungal mycelial disc (6 mm).

### 2.8. Statistical Analysis

The results of the study are presented as the mean ± standard deviation. Statistical analysis was performed using one-way analysis of variance in IBM SPSS software (Ver. 20.0, SPSS Inc., Chicago, IL, USA). Duncan’s multiple range test was used to determine significance at the 95% probability level.

## 3. Results

### 3.1. Collection of Fungal Isolates from Potato Plants with Brown Leaf Spot Disease

We investigated the occurrence of diseases affecting potatoes cultivated in three northern regions (Yeoncheon, Goseong, and Baengnyeongdo) of Korea from June and July 2020 to 2021 (Table 3). Brown leaf spot disease was observed in 2020 and 2021, with a particularly severe outbreak in Yeoncheon in 2021 (Figure 1a). The molecular identification of the isolates was performed by analysing their morphological characteristics and ITS region sequences, then cross-referencing the obtained ITS sequences with the results of the NCBI BLAST search (Table 3).

During the first round of sampling (2020), we collected 31 isolates (83.8%) of *Alternaria* spp. and six isolates (16.2%) of *Fusarium* spp. from the entire collection area. In the second round (in 2021), we collected 24 isolates (77.4%) of *Alternaria* spp., 2 isolates (6.5%) of *Fusarium* spp., 2 isolates (6.5%) of *Boeremia* spp., 2 isolates (6.5%) of *Stagonosporopsis* spp., and 1 isolate (3.2%) of *Colletotrichum* sp. from the same collection area. Altogether, the 68 isolates collected across the 2-year period comprised *Alternaria* spp. (55 isolates), *Fusarium* spp. (8 isolates), *Boeremia* spp. (2 isolates), *Stagonosporopsis* spp. (2 isolates), and *Colletotrichum* sp. (1 isolate).

### 3.2. Phylogenetic Analysis of Alternaria spp. Isolates Using Seven Barcoding Genes

Given that members of the *Alternaria* genus were the predominant fungal species detected throughout the study period, we obtained nucleotide sequences for seven barcoding genes: ITS, *gapdh*, *tef1*, *rpb2*, *Alt a 1*, *endoPG*, and OPA10-2, for species-level identification of the 55 selected *Alternaria* spp. isolates. For parts of the analysis, we used individual gene sequences (ITS (Appendix A), *gapdh* (Appendix A), *tef1* (Appendix A), *rpb2* (Appendix A), *Alt a 1* (Appendix A), *endoPG* (Appendix A), and OPA10-2 (Appendix A)). Additionally, a concatenated multigene phylogeny encompassing all seven genes was generated (Figure 2). The multigene phylogenetic tree revealed that 51 isolates corresponded to *A. alternata*, while the remaining four isolates comprised two strains of *A. arborescens* and two strains of *A. solani* (Figure 2). Notably, the phylogenetic tree generated from *rpb2* sequences showed similar clustering (Appendix A) to that seen in the concatenated seven-gene phylogenetic tree; two isolates (SYP-F0352 and SYP-F035) clustered with the type strain *A. solani* CBS109157, and two isolates (SYP-F0713 and SYP-F0714) were associated with *A. arborescens* CBS 102605 (Appendix A). However, in the remaining single-gene tree, isolates of *A. alternata*, *A. arborescens*, and *A. solani* could not be clearly distinguished.

### 3.3. Pathogenicity Test

Infections caused by *Alternaria* spp. involve the direct invasion of host plants through the stomata and/or wounds [30]. To investigate the possibility of infection through stomata, we inoculated conidia onto the entire unwounded surface of samples from the Superior potato cultivar. Based on the results of the phylogenetic analysis, the isolates were clustered into three distinct groups, and we selected isolates from each of these the three groups for virulence testing (Figure 2): six isolates (SYP-F0939, SYP-F0942, SYP-F0934, SYP-F0941, SYP-F0944, and SYP-F0946) from Group I, one isolate (SYP-F0936) from Group II, and four isolates (SYP-F0935, SYP-F0940, SYP-F0943, and SYP-F0945) from Group III.

Disease symptoms were first observed 3 days post-inoculation (dpi) in most isolates, and severe necrotic brown spot symptoms were observed at 7 dpi. Necrosis progressed from the outer edges to the inner regions of the leaves, causing them to turn black and wilt (Figure 3). These symptoms are similar to those observed in the field (Figure 1). In particular, two strains classified in Group I, SYP-F0939 and SYP-F0942, caused severe disease, leading to the death of all inoculated potato seedlings. No symptoms were observed in the control seedlings. The isolates retrieved from all diseased leaves were confirmed to be *A. alternata* based on their sequences at the *rpb2* locus, establishing *A. alternata* as the causative agent of this disease.

### 3.4. Virulence Test on Commercial Cultivars

Typical symptoms of *Alternaria* spp. infection were observed in both wounded and non-wounded leaves within 7 days of the initial exposure in the detached leaf assay. These symptoms were similar to those observed in the field. The results of the virulence test for all seven cultivars indicated that the disease incidence depended on both the potato cultivar and the *A. alternata* isolate in wounding and non-wounding inoculations. The control group showed no symptoms on either wounded or non-wounded leaves (Figure 4a).

Potato cv. Chubaek: typical progressive or acute symptoms included brownish-black lesions, leaf discoloration, and purple coloration in response to some isolates (Figure 4a). Among the seven cultivars tested, Chubaek was relatively abundant. The DLA rates for all isolates (12, 2, and 12 isolates selected from *A. alternata* groups I, II, and III, respectively, see Figure 2) were 33.2% and 52.2% with non-wounding (Figure 4b) and wounding (Figure 4c) inoculations, respectively. These results indicate the heightened susceptibility of Chubaek to the *A. alternata* isolates obtained from the potato fields, compared to the other six cultivars.

The potato cvs. Daekwang, Daeji, and Superior exhibited typical brownish-black lesions and halo formations (Figure 4a). The DLA rates for all isolates were 16.4%, 17.9%, and 19.3% for non-wounding inoculations and 19.6%, 31.0%, and 33.0% for wounding inoculations, respectively (Figure 4b).

The incidence of disease was the lowest in potato cvs. Arirang-1ho, Arirang-2ho, and Golden Ball inoculated with any of the isolates (Figure 4). The DLA rates of cvs. Arirang-1ho, Arirang-2ho, and Golden Ball were 1.2%, 2.2%, and 3.0%, respectively, for non-wounding inoculations (Figure 4a,b). For wounding inoculations (Figure 4c), the values were 11.5%, 12.3%, and 9.1%, respectively, indicating resistance to the disease. Consistent re-isolation of the pathogen from symptomatic plants of most cultivars confirmed *A. alternata* as the causative agent responsible for the observed symptoms.

### 3.5. Selection of Appropriate Fungicides for Potato Brown Spot Disease in Korea

In total, 10 isolates were used for the fungicide-selection experiment, with two selected from each of the three groups (I, II, and III) of *A. alternata* and two isolates each of *A. arborescens* and *A. solani* (Figure 2). The fungicides mancozeb and difenoconazole effectively controlled the mycelial growth of all isolates, with control rates ranging from 100% to 73.1% (Figure 5, Table 4). Pydiflumetofen and boscalid exhibited lower, yet still significant, inhibitory effects on mycelial growth compared to mancozeb and difenoconazole. In contrast, chlorothalonil, kresoxim-methyl, and thiophanate-methyl had limited effectiveness, with control of mycelial growth rates ranging from a maximum of 56% to a minimum of 3.2% (Figure 5; Table 4).

## 4. Discussion

This study aimed to identify the causal pathogen responsible for potato brown leaf spot in the three northern regions of Korea from June to August in 2020–2021, identify potato cultivars resistant to this pathogen, and select the most effective commercially available fungicides for controlling this pathogen in Korean potato crops.

Our results strongly suggest that *A. alternata* is the major pathogen responsible for brown leaf spot disease in these regions. Although *A. alternata* has long been reported as a major pathogen in potato-cultivation areas in Europe [31,32], the United States [6,10,13], China [1], Israel [5], South Africa [33], and Russia [34], it was first reported in Korea only in 2023 [3]. Subsequent research on the pathogen’s distribution, the presence of resistant varieties, and the selection of effective control agents against *A. alternata* in Korea is lacking.

The phylogenetic analysis revealed genetic disparities, showing three distinct clusters—groups I, II, and III—among *A. alternata* isolates. These groups were primarily distinguished based on the ITS region and the *rpb2* gene. In addition, the *rpb2* gene sequences were found to be highly effective in identifying strains of *A. arborescens*, *A. solani*, and *A. alternata*. Previous studies have highlighted employing a combination of gene sequences in phylogenetic analysis for distinguishing among *Alternaria* spp. [1,2,10,13,14,32], and our study accurately identified *A. alternata* using a set of gene sequences that included the ITS region, *gapdh*, *tef1*, *rpb2*, *Alt a1*, *endoPG*, and OPA 10-1. We believe that more extensive genetic information can offer deeper insights. However, despite the genotype-based clustering into groups I, II, and III, these differences did not translate into notable differences in phenotype. No significant differences were noted in pathogenicity, virulence across the seven cultivars, or sensitivity to fungicides among these genotype-based groups. This result suggests that these genetic distinctions might not be practically significant.

In the northern region of Korea, the period from late June to mid-July is a critical time for the growth and harvest of potatoes, directly impacting potential yield losses. Due to recent climate change, significant temperature fluctuations between day and night result in wet leaves and prolonged high humidity during the rainy season. These conditions may elevate the risk that potato plants will be susceptible to various pathogens, including *A. alternata*. Based on our survey, the data indicate that *A. alternata* is dominant over *A. solani* or *P. infestans* in Korea (see Table 3). In a previous study, it was noted that the occurrence of late blight decreases when the sowing date is shifted to a cooler season [35]. Therefore, it is expected that using climate-change scenarios and developing a model will enable researchers to identify actions that can be taken to decrease the prevalence of the disease.

The majority of the differences between *A. alternata* and *A. solani* or *P. infestans* may be attributed to chemical treatments. Choi et al. (2023) recently reported a shortage of fungicides developed to control *A. alternata*, the pathogen responsible for potato leaf spot disease in Korea [3]. Therefore, if a farmer sprayed fungicides on the potato field, they most likely controlled the spread of *A. solani* or *P. infestans*. Based on this information, the *A. alternata* isolated in this study is likely resistant to fungicides, especially considering that *A. alternata* was the predominant isolate. This finding is significant because both *A. solani* and *P. infestans* thrive in high humidity and at a wide range of temperatures [35,36,37]. On the other hand, it is highly likely that *A. alternata* is not resistant to mancozeb and chlorothalonil, which were not used in the fields. Therefore, testing for resistance to fungicides should be conducted. Additionally, although this issue was not investigated in our study, the extensive use of QoI fungicides has been reported to lead to resistance in anthracnose on pepper plants in Korea [38,39].

Notably, all 10 selected isolates exhibited resistance to the QoI fungicide, kresoxim-methyl (Figure 5; Table 4). Therefore, testing for resistance to other QoI fungicides, such as azoxystrobin, is warranted; the widespread use of QoIs in commercial potato fields in Korea has led to a high probability of resistance development in *Alternaria* spp. Dube et al. (2014) reported the presence of *A. alternata* isolates resistant to azoxystrobin, which is commonly used to control early blight [33]. These isolates had a single point mutation in the *cyt b* gene of the mitochondrial genome, resulting in an amino-acid substitution from glycine to alanine at position 143 (G143A) [33]. This result highlights the feasibility of confirming resistance to QoI-class pesticides through PCR amplification and sequencing. The finding of resistance emphasises the importance of continuous monitoring and management of resistant strains of *Alternaria* spp. to implement effective fungicide-application strategies.

## 5. Conclusions

This study identified *A. alternata*, *A. arborescens*, and *A. solani* as the causative agents of brown leaf spot disease in potatoes in Korea through phylogenetic analyses. Among these, *A. alternata* was found to be the major pathogen. Pathogenicity tests confirmed that all the selected isolates resulted in the same symptoms observed in the field. When inoculated with and without wounds, Arirang-1ho, Arirang-2ho, and Golden Ball exhibited resistance; Daekwang, Daeji, and Superior demonstrated moderate resistance; and Chubaek was found to be susceptible. In vitro screening identified mancozeb and difenoconazole as the most effective fungicides for inhibiting fungal growth, making them the most suitable fungicide options. Our study revealed that brown spot disease caused by *A. alternata,* which has been previously reported in various countries, has the potential to become a major disease affecting potato production in Korea. Further research is required to investigate the genetic diversity of this pathogen and environmental factors influencing the occurrence of this disease.

## Figures and Tables

**Figure 1 jof-10-00053-f001:**
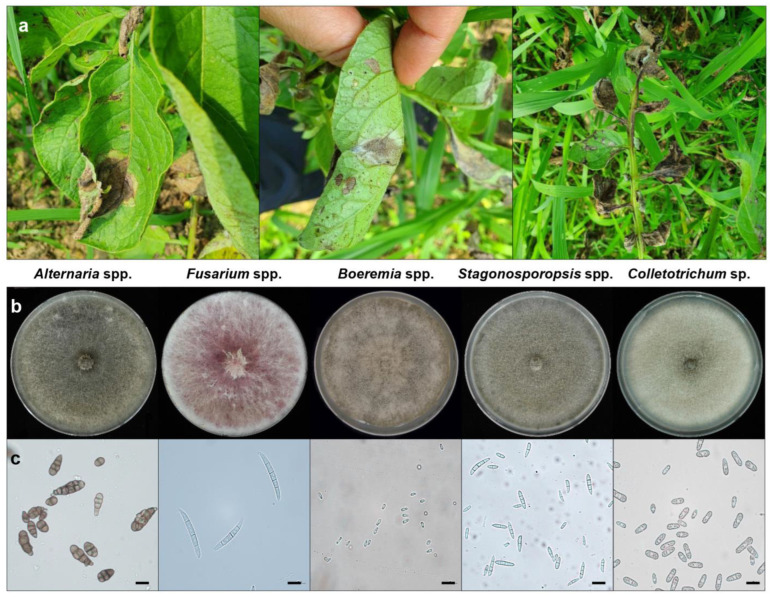
Naturally occurring leaf brown spot on potato plants, representatives of collected isolates, and percentage distribution of fungal isolates during 2020–2021 in Korea. (**a**) Brown spot symptoms on potato leaves collected from Yeoncheon (June 2021). Symptoms on the front of the leaves (left), back of the leaves (centre), and stem (right); (**b**) 14-day-old PDA cultures of five representative fungal species, including *Alternaria* spp. (far left): *Fusarium* spp. (second from the left), *Boeremia* spp. (centre), *Stagonosporopsis* spp. (right of centre), and *Colletotrichum* sp. (far right); (**c**) Spores of each fungus. The scale bar indicate 10 μm.

**Figure 2 jof-10-00053-f002:**
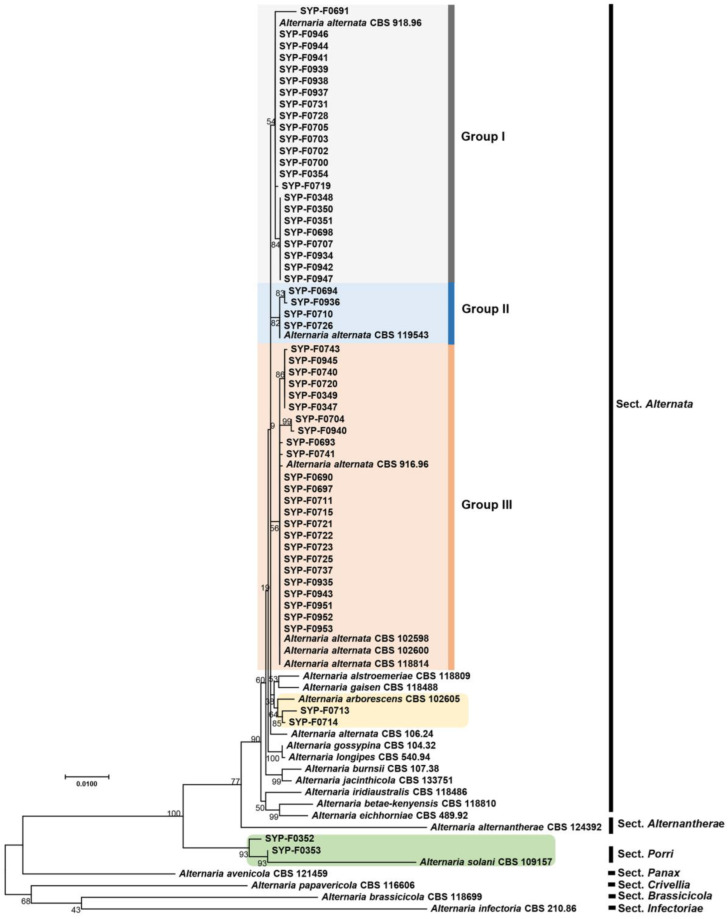
Phylogenetic analysis. Phylogenetic tree constructed based on concatenated sequences of ITS, *gapdh*, *tef1*, *rpb2*, *Alt a1*, *endoPG*, and OPA10-2 from 23 strains of *Alternaria* spp. Reference sequences were retrieved from GenBank (accession numbers shown in Appendix A). The tree was constructed using the maximum-likelihood method, and bootstrap values (1000 replications) are shown in front of each node. MEGA version X software was used for the analysis.

**Figure 3 jof-10-00053-f003:**
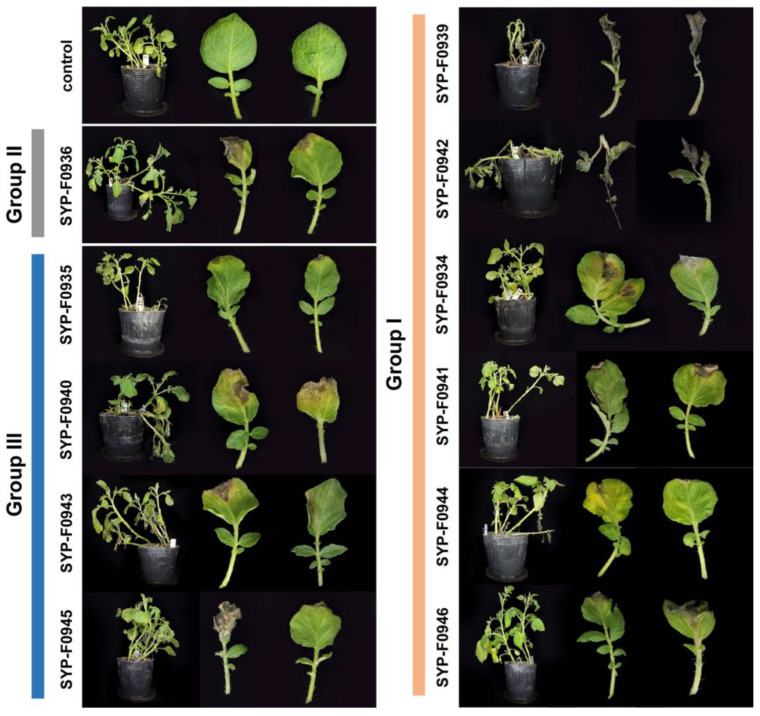
Pathogenicity test. One-month-old potato cultivar Superior plants were inoculated by spraying with conidial suspension (1 × 10^5^ conidia/mL). The photographs were captured at 7 days post-inoculation.

**Figure 4 jof-10-00053-f004:**
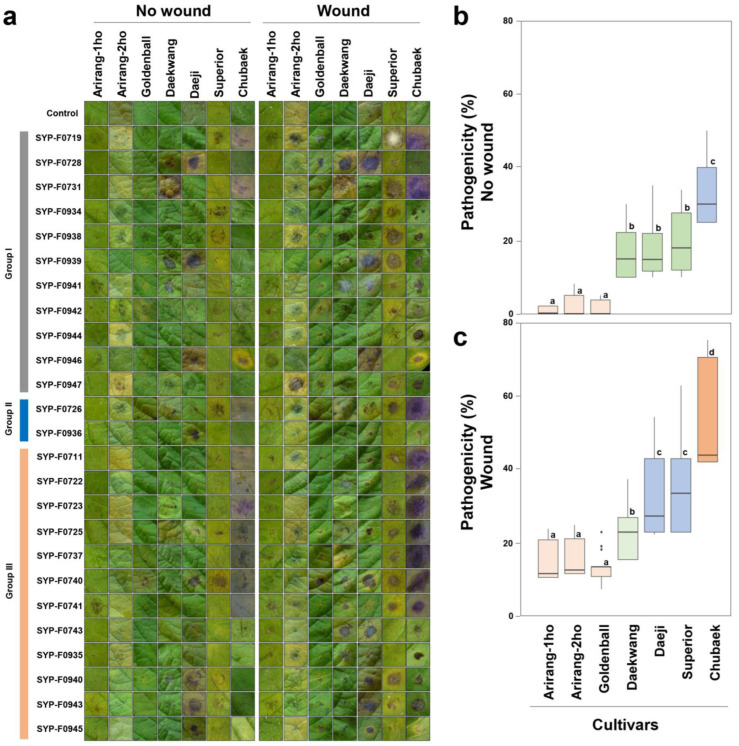
Detached leaf assay without/with wounding and box-plot analysis. (**a**) Detached leaf assay. Sterilised distilled water was used as the control. Isolates for the virulence test were selected from groups I, II, and III (on the left), which are identical to the groups shown in Figure 2. Box-plot analysis using diseased-leaf-area data from the virulence test (**b**) without wounding, and (**c**) with wounding. The use of the same colour in different box plots indicates no significant difference between cultivars. The line inside each box represents the median value. Outliers are shown as dots. Bars indicate the standard error of the means (n = 25).

**Figure 5 jof-10-00053-f005:**
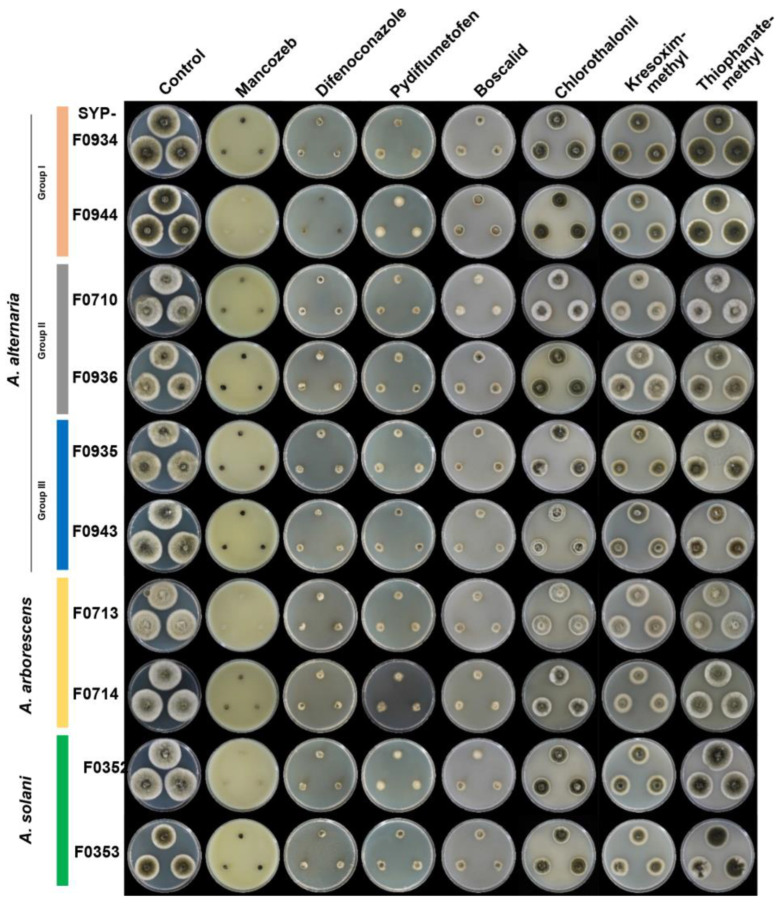
Effects of seven fungicides on growth rates of selected *Alternaria* spp. isolates causing potato brown leaf spot. Selected *Alternaria* spp. isolates grown for 7 days on PDA containing mancozeb (1500 μg/mL), chlorothalonil (1253 μg/mL), difenoconazole (34 µg/mL), pydiflumetofen (46 μg/mL), boscalid (328 μg/mL), kresoxim-methyl (148 μg/mL), or thiophanate-methyl (700 μg/mL).

**Table 1 jof-10-00053-t001:** Primers used for PCR and sequencing.

Locus ^a^	Primer	Primer Sequence (5′-3′)	References
ITS	V9G	TTACGTCCCTGCCCTTTGTA	[16]
	ITS4	CCTCCGCTTATTGATATGC	[17]
*gapdh*	gpd1	CAACGGCTTCGGTCGCATTG	[18]
	gpd2	GCCAAGCAGTTGGTTGTGC	[18]
*tef1*	EF1-728F	CATCGAGAAGTTCGAGAAGG	[19]
	EF1-986R	TAC TTG AAG GAA CCC TTA CC	[19]
	EF2	GGARGTACCAGTSATCATGTT	[20]
*rpb2*	RPB2-5F2	GGGGWGAYCAGAAGAAGGC	[21]
	fRPB2-7cR	CCCATRGCTTGTYYRCCCAT	[22]
*Alt a 1*	Alt-For	ATGCAGTTCACCACCATCGC	[23]
	Alt-Rev	ACGAGGGTGAYGTAGGCGTC	[23]
*endoPG*	PG3	TACCATGGTTCTTTCCGA	[24]
	PG2b	GAGAATTCRCARTCRTCYTGRTT	[24]
OPA10-2	OPA 10-2R	GATTCGCAGCAGGGAAACTA	[24]
	OPA 10-2L	TCGCAGTAAGACACA TTCTACG	[24]

^a^ ITS: internal transcribed spacer regions 1 and 2 and intervening 5.8S nrDNA, *gapdh*: glyceraldehyde 3-phosphate dehydrogenase, *tef1*: translation elongation factor 1-alpha, *rpb2*: RNA polymerase second largest subunit, *Alt a 1*: *Alternaria* major allergen gene, *endoPG*: endopolygalacturonase, and OPA10-2: anonymous region.

**Table 2 jof-10-00053-t002:** Chemical names, group name, formulation, and final concentration in the medium used in the *Alternaria* trials.

Chemical Name	Target Site	Group Name	Manufacturer	Formulation (%)	Concentration Recommended by the Manufacturer	Final Conc. of Fungicide in the Medium (μg/mL)
Mancozeb	Multi-site contact activity	Dithiocarbamates	Daeyu Co., Ltd., Seoul, Republic of Korea	75	500-fold	1500
Chlorothalonil		Chloronitriles	Hanearl Science Ltd., Seoul, Republic of Korea	75	600-fold	1253
Difenoconazole	Inhibit sterol biosynthesis in membrane	C14-methylase in sterol biosynthesis	Hanearl Science Ltd., Seoul, Republic of Korea	10	3000-fold	34
Pydiflumetofen	“Complex II” Succinate dehydrogenase	Succinate-dehydrogenase inhibitor (SDHI)	Agrigento Co., Kyungnam, Republic of Korea	18.35	1500-fold	46
Boscalid			Syngenta Republic of Korea, Seoul, Republic of Korea	49.30	4000-fold	328
Krexosim-methyl	Inhibit mitochondrial respiration	Quinone outside inhibitor (QoI)	Nonghyup chemical, Seoul, Republic of Korea	40.20	3000-fold	148
Thiophanate-methyl	B1 tubulin polymerization	Methyl benzimidazole carbamates (MBC)	FarmHannong, Ltd., Seoul, Republic of Korea	70	1000-fold	700

**Table 3 jof-10-00053-t003:** List of collected isolates, localities, species, and GenBank accession numbers for ITS region sequences.

Isolates	The Closest Matching GenBank Taxa	GenBank Accession Nos.	Query Over	Percent	Collected Regions	Date of Isolation
SYP-F0690	*Alternaria alternata*	OR787445.1	100	100	Goseong	26 June 2020
SYP-F0691	*Alternaria alternata*	OR787445.1	100	100	Goseong	26 June 2020
SYP-F0693	*Alternaria alternata*	OR787445.1	100	100	Goseong	26 June 2020
SYP-F0694	*Alternaria alternata*	OR787445.1	100	100	Goseong	26 June 2020
SYP-F0697	*Alternaria alternata*	OR787445.1	100	100	Goseong	26 June 2020
SYP-F0698	*Alternaria alternata*	OR787445.1	100	100	Goseong	26 June 2020
SYP-F0700	*Alternaria alternata*	OR787445.1	100	100	Goseong	26 June 2020
SYP-F0702	*Alternaria alternata*	OR787445.1	100	100	Goseong	26 June 2020
SYP-F0703	*Alternaria alternata*	OR787445.1	100	100	Goseong	26 June 2020
SYP-F0704	*Alternaria alternata*	OR787445.1	100	100	Goseong	26 June 2020
SYP-F0705	*Alternaria alternata*	OR787445.1	100	100	Goseong	26 June 2020
SYP-F0707	*Alternaria alternata*	OR787445.1	100	100	Goseong	26 June 2020
SYP-F0696	*Fusarium acuminatum*	MT635295.1	100	100	Goseong	26 June 2020
SYP-F0688	*Fusarium equiseti*	MT560375.1	100	100	Goseong	26 June 2020
SYP-F0689	*Fusarium equiseti*	MT560375.1	100	100	Goseong	26 June 2020
SYP-F0692	*Fusarium equiseti*	MT560375.1	100	100	Goseong	26 June 2020
SYP-F0708	*Fusarium equiseti*	MT560375.1	100	100	Goseong	26 June 2020
SYP-F0709	*Fusarium equiseti*	MT560375.1	100	100	Goseong	26 June 2020
SYP-F0347	*Alternaria alternata*	MH992147.1	100	100	Goseong	22 July 2020
SYP-F0348	*Alternaria alternata*	OR787445.1	100	100	Goseong	22 July 2020
SYP-F0349	*Alternaria alternata*	KX816031.1	100	100	Goseong	22 July 2020
SYP-F0350	*Alternaria alternata*	OR787445.1	100	99.81	Goseong	22 July 2020
SYP-F0351	*Alternaria alternata*	OR787445.1	100	100	Goseong	22 July 2020
SYP-F0354	*Alternaria alternata*	OR787445.1	100	100	Baengnyeongdo	22 July 2020
SYP-F0352	*Alternaria solani*	MT498268.1	100	100	Baengnyeongdo	22 July 2020
SYP-F0353	*Alternaria solani*	OR787445.1	100	100	Baengnyeongdo	22 July 2020
SYP-F0937	*Alternaria alternata*	OR787445.1	100	100	Yeoncheon	27 June 2020
SYP-F0938	*Alternaria alternata*	OR787445.1	100	100	Yeoncheon	27 June 2020
SYP-F0939	*Alternaria alternata*	OR787445.1	100	100	Yeoncheon	27 June 2020
SYP-F0940	*Alternaria alternata*	OR787445.1	100	100	Yeoncheon	27 June 2020
SYP-F0941	*Alternaria alternata*	OR787445.1	100	100	Yeoncheon	27 June 2020
SYP-F0942	*Alternaria alternata*	OR787445.1	100	100	Yeoncheon	27 June 2020
SYP-F0943	*Alternaria alternata*	OR787445.1	100	100	Yeoncheon	27 June 2020
SYP-F0944	*Alternaria alternata*	OR787445.1	100	100	Yeoncheon	27 June 2020
SYP-F0945	*Alternaria alternata*	OR787445.1	100	100	Yeoncheon	27 June 2020
SYP-F0946	*Alternaria alternata*	OR787445.1	100	100	Yeoncheon	27 June 2020
SYP-F0947	*Alternaria alternata*	OR787445.1	100	100	Yeoncheon	27 June 2020
SYP-F0710	*Alternaria alternata*	OR787445.1	100	100	Yeoncheon	22 June 2021
SYP-F0711	*Alternaria alternata*	OR787445.1	100	100	Yeoncheon	22 June 2021
SYP-F0715	*Alternaria alternata*	OR687203.1	100	99.61	Yeoncheon	22 June 2021
SYP-F0719	*Alternaria alternata*	OR787445.1	100	100	Yeoncheon	22 June 2021
SYP-F0720	*Alternaria alternata*	OR787445.1	100	100	Yeoncheon	22 June 2021
SYP-F0721	*Alternaria alternata*	OR734592.1	100	100	Yeoncheon	22 June 2021
SYP-F0722	*Alternaria alternata*	OR787445.1	100	100	Yeoncheon	22 June 2021
SYP-F0723	*Alternaria alternata*	OR787445.1	100	100	Yeoncheon	22 June 2021
SYP-F0725	*Alternaria alternata*	OR787445.1	100	100	Yeoncheon	22 June 2021
SYP-F0726	*Alternaria alternata*	OR787445.1	100	100	Yeoncheon	22 June 2021
SYP-F0728	*Alternaria alternata*	OR787445.1	100	100	Yeoncheon	22 June 2021
SYP-F0731	*Alternaria alternata*	OR787445.1	100	100	Yeoncheon	22 June 2021
SYP-F0737	*Alternaria alternata*	OR787445.1	100	100	Yeoncheon	22 June 2021
SYP-F0740	*Alternaria alternata*	OR787445.1	100	100	Yeoncheon	22 June 2021
SYP-F0741	*Alternaria alternata*	ON599295.1	100	98.33	Yeoncheon	22 June 2021
SYP-F0743	*Alternaria alternata*	MT498268.1	100	100	Yeoncheon	22 June 2021
SYP-F0934	*Alternaria alternata*	OR787445.1	100	100	Yeoncheon	22 June 2021
SYP-F0935	*Alternaria alternata*	OR787445.1	100	100	Yeoncheon	22 June 2021
SYP-F0936	*Alternaria alternata*	OR787445.1	100	100	Yeoncheon	22 June 2021
SYP-F0713	*Alternaria arborescens*	MT212228.1	100	100	Yeoncheon	22 June 2021
SYP-F0714	*Alternaria arborescens*	OR787445.1	100	100	Yeoncheon	22 June 2021
SYP-F0736	*Boeremia exigua*	KY555024.1	100	100	Yeoncheon	22 June 2021
SYP-F0733	*Boeremia exigua*	MT397284.1	100	100	Yeoncheon	22 June 2021
SYP-F0730	*Colletotrichum nymphaeae*	LC435466.1	100	100	Yeoncheon	22 June 2021
SYP-F0732	*Fusarium equiseti*	MK752407.1	100	100	Yeoncheon	22 June 2021
SYP-F0734	*Fusarium graminearum*	OR346117.1	100	100	Yeoncheon	22 June 2021
SYP-F0724	*Stagonosporopsis dennisii*	OQ158929.1	100	99.18	Yeoncheon	22 June 2021
SYP-F0727	*Stagonosporopsis dennisii*	OK315470.1	100	100	Yeoncheon	22 June 2021
SYP-F0951	*Alternaria alternata*	OR787445.1	100	100	Baengnyeongdo	19 July 2021
SYP-F0952	*Alternaria alternata*	OR787445.1	100	100	Baengnyeongdo	1 August 2021
SYP-F0953	*Alternaria alternata*	OR787445.1	100	100	Baengnyeongdo	1 August 2021

**Table 4 jof-10-00053-t004:** Rates of inhibition of fungal growth by seven selected fungicides tested against *Alternaria* spp.

Isolates	Control	Mancozeb	Difenoconazole	Pydiflumetofen	Boscalid	Chlorothalonil	Kresoxim-Methyl	Thiophanate-Methyl
SYP-F0934	0.0 ± 0.0 ^g,#^	100.0 ± 0.0 ^a^	90.0 ± 1.6 ^b^	77.3 ± 4.2 ^c^	77.3 ± 1.6 ^c^	46.4 ± 1.6 ^d^	38.2 ± 3.1 ^e^	14.6 ± 3.1 ^f^
SYP-F0944	0.0 ± 0.0 ^g^	100.0 ± 0.0 ^a^	96.3 ± 1.6 ^b^	75.7 ± 1.6 ^c^	72.9 ± 1.6 ^c^	58.0 ± 2.8 ^d^	39.3 ± 3.2 ^e^	20.6 ± 1.6 ^f^
SYP-F0945	0.0 ± 0.0 ^f^	100.0 ± 0.0 ^a^	90.5 ± 1.6 ^b^	84.8 ± 3.3 ^c^	81.0 ± 1.6 ^c^	41.9 ± 1.6 ^d^	44.8 ± 4.4 ^d^	25.7 ± 2.9 ^e^
SYP-F0936	0.0 ± 0.0 ^g^	100.0 ± 0.0 ^a^	87.3 ± 2.5 ^b^	75.8 ± 1.9 ^c^	68.1 ± 1.9 ^d^	40.0 ± 1.0 ^e^	3.2 ± 1.9 ^g^	10.9 ± 1.7 ^f^
SYP-F0935	0.0 ± 0.0 ^f^	100.0 ± 0.0 ^a^	83.7 ± 2.7 ^b^	77.3 ± 1.6 ^bc^	80.0 ± 5.7 ^c^	51.9 ± 1.6 ^d^	55.5 ± 1.6 ^d^	28.2 ± 1.6 ^e^
SYP-F0943	0.0 ± 0.0 ^f^	100.0 ± 0.0 ^a^	92.6 ± 1.6 ^b^	82.4 ± 4.2 ^c^	80.6 ± 2.8 ^c^	56.5 ± 3.2 ^d^	50.0 ± 2.8 ^e^	45.4 ± 4.2 ^e^
SYP-F0713	0.0 ± 0.0 ^h^	100.0 ± 0.0 ^a^	88.1 ± 0.0 ^c^	97.0 ± 0.0 ^b^	74.3 ± 1.7 ^d^	41.6 ± 1.7 ^e^	32.7 ± 3.4 ^f^	13.9 ± 0.0 ^g^
SYP-F0714	0.0 ± 0.0 ^f^	100.0 ± 0.0 ^a^	90.0 ± 8.7 ^b^	76.0 ± 3.0 ^c^	80.0 ± 1.7 ^c^	44.9 ± 1.7 ^d^	43.9 ± 1.7 ^d^	14.9 ± 1.7 ^e^
SYP-F0352	0.0 ± 0.0 ^g^	100.0 ± 0.0 ^a^	89.0 ± 2.8 ^b^	79.8 ± 1.6 ^c^	79.8 ± 1.6 ^c^	52.2 ± 1.6 ^d^	43.1 ± 1.6 ^e^	14.6 ± 7.3 ^f^
SYP-F0353	0.0 ± 0.0 ^f^	98.7 ± 2.2 ^a^	73.1 ± 20.0 ^b^	74.4 ± 2.2 ^b^	66.7 ± 2.2 ^b^	37.2 ± 2.2 ^c^	25.6 ± 2.2 ^c^	5.1 ± 2.2 ^d^

Data are presented as mean ± SD from three replicates. ^#^ Duncan’s multiple range test was used to determine significance at the 95% probability level. The presence of the same letters multiple times in a column indicates no significant difference between those results.

## Data Availability

Data are contained within the article and Appendix A.

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
