# Peer review of "Isolation and Identification of Alternaria alternata from Potato Plants Affected by Leaf Spot Disease in Korea: Selection of Effective Fungicides"

_jof, 2024, doi:10.3390/jof10010053_

Round 1

Reviewer 1 Report

Comments and Suggestions for Authors

Manuscript ID: jof-2800159. Isolation and Identification of Alternaria alternata from Potato Leaf Spot Disease in Korea: Selection of Effective Fungicides.

The manuscript is well written and this reviewer has only minor corrections, comments and suggestions.

L49-50: Which environmental conditions are favourable for the pathogen in Korea? Please specify.

Do the authors know how detrimental (yield loss) the brown spot disease is compared to early and late blight in Korea?

L63: “fungal species” instead of “fungi species”.

L101: Which brand of V8-juice was used? What was used to adjust pH? Please specify.

L146-155: Koch’s postulates need to be described somewhere around this paragraph.

L183-188: Delete the parentheses after each fungicide in the text and keep the information in Table 2.

Figure 1 - Suggestion: delete the pie charts (the information is already in the text) and enlarge a to c to fill the full-page width. It is more interesting to see the spots, the colonies and the conidia that the charts.

Figure 3 - Suggestion: enlarge the figure to fill the full-page width. It would also make it possible to compare spots in fig. 1 with the spots in fig. 3.

Figure 4 – legend: add a sentence that state that says something like “the same colour in the box plots show/indicate no significant difference”.

Figure 5 same as for figures 1 and 3.

General discussion/perspectives: It would be interesting if the authors could discuss/speculate on potato growing in Korea in respect to (1) climate changes and (2) fungicide resistance by the fungi.

Will it be possible grow a reasonable crop of potato in the future? Would it be to dry or to wet and therefore bad for the potato yield in the long run. Would a high use of fungicides be a valid way to go? Azole use in Europe have made Aspergillus fumigatus resistance and very difficult to tread aspergillosis.

Author Response

Dear reviewer,

We are grateful for the chance to improve our manuscript with resubmission. We have fully revised the manuscript. Our responses are highlighted in blue.

We hope this revised manuscript is acceptable for publication in Journal of Fungi. If you need any additional information or recommend any additional changes, please let me know. Once again, we appreciate your effort to improve the quality of our manuscript.

Sincerely,

Sook-Young Park

Reviewer 1

  1. L49-50: Which environmental conditions are favourable for the pathogen in Korea? Please specify.

-- We added the information for environmental conditions.

  1. Do the authors know how detrimental (yield loss) the brown spot disease is compared to early and late blight in Korea?

-- Unfortunately, the yield loss caused by the brown spot disease, as well as early and late blight, was not accurately calculated in Korea. Thus, we cannot include this information.

  1. L63: “fungal species” instead of “fungi species”.

-- We changed “fungi species” to “fungal species”.

  1. L101: Which brand of V8-juice was used? What was used to adjust pH? Please specify.

-- We changed “8% V8 Juice, 1.5% agar” to “8% Campbell’s V8 Juice, 1.5% agar, pH adjusted to 7 using 0.1N NaOH”.

  1. L146-155: Koch’s postulates need to be described somewhere around this paragraph.

-- We changed “To test for pathogenicity” to “To fulfill Koch’s postulates and confirm that these fungi could infect potato leaves”.

  1. L183-188: Delete the parentheses after each fungicide in the text and keep the information in Table 2.

-- We deleted the parentheses.

  1. Figure 1 - Suggestion: delete the pie charts (the information is already in the text) and enlarge a to c to fill the full-page width. It is more interesting to see the spots, the colonies and the conidia that the charts.

-- We deleted the pie charts in Figure 1.

  1. Figure 3 - Suggestion: enlarge the figure to fill the full-page width. It would also make it possible to compare spots in fig. 1 with the spots in fig. 3.

-- We deleted the pie charts in Figure 1.

  1. Figure 4 – legend: add a sentence that state that says something like “the same colour in the box plots show/indicate no significant difference”.

-- We added the information to the legend of Figure 4.

  1. Figure 5 same as for figures 1 and 3.

-- We modified.

  1. General discussion/perspectives: It would be interesting if the authors could discuss/speculate on potato growing in Korea in respect to (1) climate changes and (2) fungicide resistance by the fungi.

Will it be possible grow a reasonable crop of potato in the future? Would it be to dry or to wet and therefore bad for the potato yield in the long run. Would a high use of fungicides be a valid way to go? Azole use in Europe have made Aspergillus fumigatus resistance and very difficult to tread aspergillosis.

-- We modified in Discussion.

Reviewer 2 Report

Comments and Suggestions for Authors

Dear Authors!

The manuscript is devoted to the actual topic of combating diseases of food crops. The manuscript will certainly arouse the interest of readers, but in order to improve its quality, I propose to make some changes to it.

1. One of the goals of the work was “(i) to identify the predominant causal pathogen... via morphological and molecular analyses...”. The authors performed molecular analysis, but did not study the morphological characteristics of isolates of phytopathogenic micromycetes.

2. Another goal of the study was “(iii) to examine their pathogenicity on different potato cultivars.” However, the pathogenicity test was carried out only on Superior potatoes. Why hasn't research been done on other varieties?

3. Table 3 shows 77 fungal isolates, of which 64 belong to the genus Alternaria. Therefore, it is not clear why 55 isolates of Alternaria spp. were selected for phylogenetic analysis. (line 253).

4. Fig. 4c. It is questionable that there are no significant differences between the Arirang-1ho and Arirang-2ho cultivars, on the one hand, and the Golden Ball cultivar, on the other hand.

5. In the Materials and Methods section, indicate the manufacturers of fungicides and explain the choice of their concentrations, which are indicated in Fig. 5.

6. Conclusions need correction, because the authors did not perform morphological analysis to identify pathogens.

Comments on the Quality of English Language

 Minor editing of English language required

Author Response

Dear reviewer,

We are grateful for the chance to improve our manuscript with resubmission. We have fully revised the manuscript. Our responses are highlighted in blue.

We hope this revised manuscript is acceptable for publication in Journal of Fungi. If you need any additional information or recommend any additional changes, please let me know. Once again, we appreciate your effort to improve the quality of our manuscript.

Sincerely,

Sook-Young Park

Reviewer 2

  1. One of the goals of the work was “(i) to identify the predominant causal pathogen... via morphological and molecular analyses...”. The authors performed molecular analysis, but did not study the morphological characteristics of isolates of phytopathogenic micromycetes.

-- We removed “morphological and” from line 84.

  1. Another goal of the study was “(iii) to examine their pathogenicity on different potato cultivars.” However, the pathogenicity test was carried out only on Superior potatoes. Why hasn't research been done on other varieties?

-- We collected the leaf spot disease from the cultivar 'Superior'. This information has been added to Line 96-97. Thus, we initially conducted pathogenicity tests on ‘Superior’ potatoes, followed by tests on 6 other cultivars including ‘Superior’.

  1. Table 3 shows 77 fungal isolates, of which 64 belong to the genus Alternaria. Therefore, it is not clear why 55 isolates of Alternaria spp. were selected for phylogenetic analysis. (line 253).

-- To avoid confusion, we excluded 9 Alternaria isolates from Table 3 because they had only undergone analysis of the DNA sequence about ITS region. Additionally, we have described the modified content.

  1. Fig. 4c. It is questionable that there are no significant differences between the Arirang-1ho and Arirang-2ho cultivars, on the one hand, and the Golden Ball cultivar, on the other hand.

-- First of all, I am very grateful to the reviewer. We have confirmed that there was a slight mistake in generating the box plot (Figure 4c). Hence, we have re-generated the box plot for Figure 4c.

  1. In the Materials and Methods section, indicate the manufacturers of fungicides and explain the choice of their concentrations, which are indicated in Fig. 5.

-- We have included the information, such as manufacturers and recommended concentrations of the chemicals, in Table 2.

  1. Conclusions need correction, because the authors did not perform morphological analysis to identify pathogens.

-- We removed “morphological” from conclusions.